# Meeting the Moment: Leveraging Temporal Inequality for Temporal Targeting to Achieve Water-Quality Load-Reduction Goals

**Nicole Opalinski [1], Daniel Schultz [1], Tamie L. Veith [2], Matt Royer [3,4] and Heather E. Preisendanz [1,4,*]**

[1] Department of Agricultural and Biological Engineering, The Pennsylvania State University, State College, PA 16802, USA; nicoleopalinski@gmail.com (N.O.); dp26schultz@gmail.com (D.S.)

[2] Pasture Systems and Watershed Management Research Unit, United States Department of Agriculture-Agricultural Research Service, State College, PA 16802, USA; tamie.veith@usda.gov

[3] Department of Agricultural Economics, Sociology and Education, The Pennsylvania State University, State College, PA 16802, USA; mzr154@psu.edu

[4] Institute for Sustainable Agriculture, Food and Environmental Science, The Pennsylvania State University, State College, PA 16802, USA

\* Correspondence: hpreisen@psu.edu; Tel.: +1-814-863-1817

**Abstract:** Inequality is an emergent property of complex systems. In catchments, variation in hydroclimatic conditions and biogeochemistry cause streamflow and constituent loads to exhibit strong temporal inequality, with most loads exported during "hot moments". Achieving water-quality-restoration goals in a cost-effective manner requires targeted implementation of conservation practices in "hot spots" in the landscape and "hot moments" in time. While spatial targeting is commonly included in development of watershed management plans, the need for temporal targeting is often acknowledged, but no common way to address it has been established. Here, we implement a Lorenz Inequality decision-making framework that uses Lorenz Curves and Gini Coefficients to quantify the degree of temporal inequality exhibited by contaminant loads and demonstrate its utility for eight impaired catchments in the Chesapeake Bay watershed. The framework requires a load-reduction goal be set and then links the degree of temporal inequality in annual nutrient loads to the periods of time during which those loads could be targeted. These results are critical in guiding development of site-specific, cost-effective tools that facilitate load-reduction and water-quality goal attainment for individual catchments. The framework provides valuable insight into site-specific potentials for meeting load-reduction goals.

**Keywords:** conservation practices; decision making; nutrients; sediment; targeting; water quality



## 1. Introduction

Inequality is a ubiquitous, emerging property of complex systems. In catchments, the spatial and temporal inequality of hydrologic and biogeochemical responses lead to the emergence of "hot spots" and "hot moments", with the vast majority of these responses occurring during relatively short periods of time and in relatively small locations. While the importance of spatial and temporal inequality is widely recognized, the methods used to identify "hot spots" and "hot moments" are not well established, with the methodology employed to analyze spatial data generally disconnected and inconsistent with the methodology employed to analyze temporal data.

The quantification of "hot spots" has been more consistently reported in the literature than the quantification of "hot moments". By calculating area-normalized loads (or other nutrient-cycle responses, such as gaseous emissions), "hot spots" are identified as the locations over a given spatial extent of interest (i.e., field, catchment, or watershed) that have the highest loads per unit area. If an area of interest needs to be managed for water-quality impairment, for example, then decision makers can direct resources to a

relatively small number of places, knowing that implementing conservation practices in those locations will achieve a higher impact on load reduction than placing the same resources and practices in other areas. Crop nutritionists increasingly support this principle through management framework called "4R Nutrient Stewardship": right nutrient source at the right rate at the right time in the right place [1]. Previous research has shown that spatially targeting adoption of agricultural conservation practices at the field scale leads to larger load-reduction goals at the watershed scale [2–6]. It is also important to recognize and manage temporal inequalities, or "hot moments", such that resources can be targeted based on both spatial and temporal inequalities. However, no uniform metric for describing temporal inequality has been widely adopted despite the prevalence of temporal inequality documentation across small and large watersheds [7–10].

The need to quantify the degree of inequality in a system is not new. Perhaps nowhere has the degree of inequality been more routinely quantified than in economics. For more than a century, Lorenz Inequality and the corresponding Gini Coefficient ($G$) have been used to determine wealth distribution by quantifying the degree of income inequality in a population. Lorenz Inequality analysis was first applied to quantify the degree of inequality for streamflow hydrology and water quality in 22 locations in the Lake Okeechobee watershed [11] and has since been utilized globally at the continental scale to better understand how climate change is likely to affect flow regimes [12]. Additionally, the analysis has been applied to time series data for geogenic constituents, nutrients, sediment, and pesticides in more than 100 watersheds ranging from 2.5 km$^2$ to 70,000 km$^2$ and at time scales ranging from daily to annually [13–15].

Water-quality degradation of coastal water bodies due to the presence of excess nutrients is a leading global environmental concern [16], with agricultural activities identified as common contributors to degraded water quality [17]. The Chesapeake Bay is the third largest estuary in the world and has a watershed area spanning 166,000 km$^2$ across seven jurisdictions. In 2010, a federally mandated Total Maximum Daily Load (TMDL) was established by the United States Environmental Protection Agency, designed to reduce nutrient and sediment loads and restore water quality to be in compliance with the Bay's designated use of fishing and swimming by 2025 [18]. To achieve mandated load-reduction goals, widespread adoption of conservation practices has occurred across the Chesapeake Bay watershed. However, current Chesapeake Assessment Scenario Tool (CAST) estimates of load reductions indicate that the Commonwealth of Pennsylvania (PA) in particular is behind the pace likely needed to meet the 2025 reduction goals [19]. Although a range of factors contribute to the overall water quality of the Chesapeake Bay, we argue that a failure to target load reduction during "hot moments" is a contributing factor. The Commonwealth of PA has established a four-tiered system for prioritizing spatial adoption of conservation practices, with each tier of counties needing to reduce 25% of the state's portion of the overall Chesapeake Bay TMDL in its current Watershed Implementation Plan [20]. Tier 1 consists of the two greatest "hot spot" counties that rank highest in nutrient and sediment loads, Tier 2 consists of five counties, whereas Tiers 3 and 4 consist of 16 and 20 counties, respectively. However, no efforts have been documented towards effectively target "hot moments".

The goal of this study is to demonstrate the impact that temporal variability from year to year can have on achieving load-reduction goals in an impaired watershed through the development of a decision-making framework for temporal targeting of "hot moments" during which the targeted load is exported. The framework consists of a novel application of Lorenz Inequality to link the temporal inequality of contaminant loads to the specific "windows of opportunity" and corresponding flow conditions necessary to target to achieve the desired load-reduction goals. The framework is demonstrated here using daily load and discharge data from eight impaired catchments in the Chesapeake Bay watershed. By comparing these loads on an annual basis with established load-reduction goals for each catchment, we determine the catchment-specific variability in percent reduction

needed from year to year and discuss how this framework enables watershed planners to understand and inform stakeholders of the risk of a watershed conservation plan.

## 2. Materials and Methods

### 2.1. Study Site Selection

The Chesapeake Bay Nontidal Network is a monitoring network comprising 123 water-quality-monitoring stations throughout the Chesapeake Bay watershed that provide nutrient and sediment data [21]. The water-quality-monitoring stations are co-located with U.S. Geological Survey (USGS) streamflow gauges, allowing loads to be calculated. While streamflow data were collected at the sub-daily scale, water-quality data were collected monthly and during targeted storm events, providing 20 data points per station per year. Load data are available in the USGS database at monthly and annual time scales back to 1985; however, daily-scale data were estimated by USGS using the weighted regression on time, discharge, and season (WRTDS) load-estimation technique [21,22].

Here, we analyze eight stations from the Chesapeake Bay Nontidal Network (Table 1; Figure 1). These stations were selected because they are located in Tier 1 or Tier 2 counties in PA, which are the counties with the greatest nutrient loads to the Chesapeake Bay watershed [20]. Further, the drainage areas for each of these monitoring stations are within one county's boundaries, enabling more accurate calculation of the specific load-reduction goals that need to be met at each point (Table 1). The load reduction for TN and TP that each county needs to meet is specified in the PA Watershed Implementation Plan [20], and the load reduction needed for each selected study site was calculated based on the size of the drainage area relative to the county. For example, a drainage area that spans half of a county would need to meet half of the county's mandated load reduction.

**Table 1.** Information regarding drainage area, percentage of land use (agricultural, forested, and urban), and annual load-reduction goals for total nitrogen (TN) and total phosphorus (TP) for each selected study site in the Pennsylvania portion of the Chesapeake Bay watershed. Land use is based on USGS 2016 National Land Cover Data (https://www.mrlc.gov/data/nlcd-2016-land-cover-conus; accessed on 30 June 2020).

| Station ID | Stream Name | County | Drainage Area (km$^2$) | Land Use (%) Forested/ Developed/ Agriculture | TN Load Reduction (kg-N/y) | TP Load Reduction (kg-P/y) |
|---|---|---|---|---|---|---|
| 1570000 | Conodoguinet Creek | Cumberland | 1217.29 | 39/20/24 | 847,735 | 11,439 |
| 1573160 | Quittapahilla Creek | Lebanon | 192.18 | 15/33/51 | 204,356 | 6916 |
| 1573695 | Conewago Creek | Lebanon | 53.09 | 42/15/41 | 56,454 | 1910 |
| 1574000 | West Conewago Creek | York | 1320.89 | 57/16/26 | 1,016,787 | - |
| 1575585 | Codorus Creek | York | 691.53 | 37/34/28 | 532,322 | - |
| 1576754 | Conestoga River | Lancaster | 1217.29 | 17/42/39 | 2,483,862 | 101,460 |
| 1576787 | Pequea Creek | Lancaster | 383.32 | 48/13/39 | 782,158 | 31,949 |
| 1614500 | Conococheague | Franklin | 1279.45 | 19/16/64 | 839,974 | 28,985 |

-, Mandated load-reduction goal has already been met.

### 2.2. Lorenz Inequality Analysis

The extent of temporal inequality in the TN and TP loads observed at each selected monitoring station were determined using Lorenz Curves and corresponding Gini coefficients (*G*). The Lorenz Curves were created by sorting the daily loads in ascending order and graphing the fractions of the cumulative loads as a function of the fractions of cumulative time (Figure 2). Lorenz Curves were generated for each station from 2010 through 2018 given that the TMDL began in 2010 and data are available from the Chesapeake Bay Nontidal Network through 2018. In a few cases, where data were not available from 2010–2012, data were analyzed from 2013 through 2018.

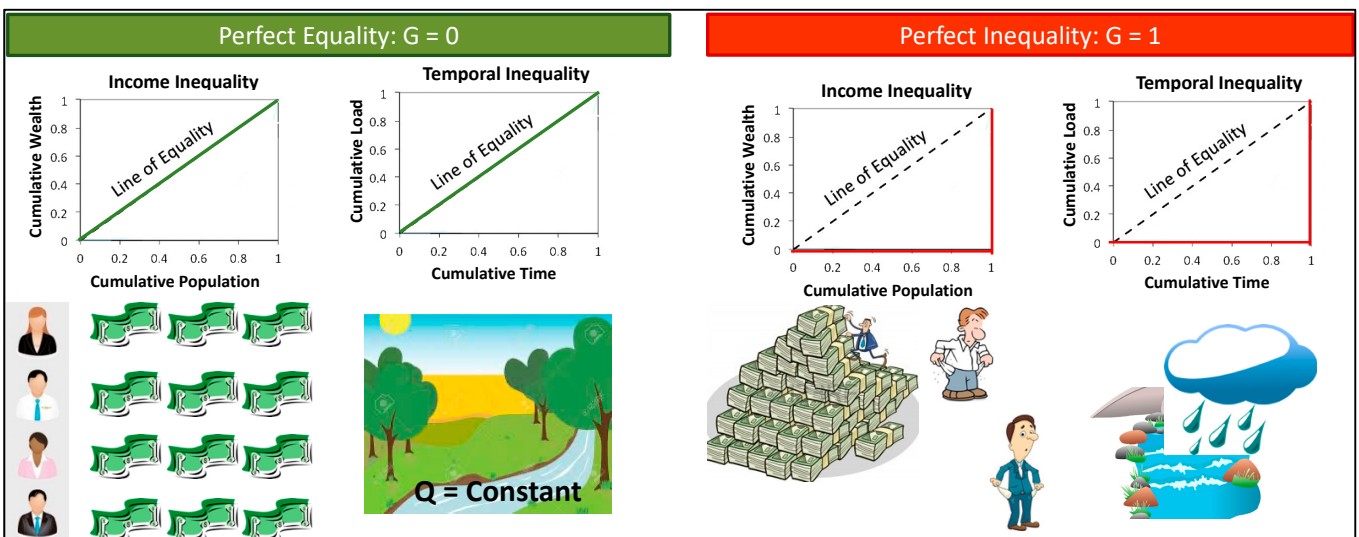

**Figure 1.** Land-use composition of the eight selected study site in the Pennsylvania portion of the Chesapeake Bay watershed. Base layer for the maps is the USGS 2016 National Land Cover Data (Available at: https://www.mrlc.gov/national-land-cover-database-nlcd-2016; accessed on 30 June 2020).

**Figure 2.** Visual representation of Lorenz Curves for perfect equality (*G* = 0) and perfect inequality (*G* = 1) scenarios in economics and their hydrologic analogues, where *G* is the Gini Coefficient.

Lorenz Curves plot on or below a line of equality (Figure 2), with the value of *G* quantifying the extent to which the curve plots below the line of equality. *G* was calculated as the ratio of the area between the line of equality and the Lorenz Curve to the entire area under the line of equality (Figure 2). These metrics are commonly used in economics and have more recently been applied to hydrology and water-quality data [11–15]. In the economics scenario, *G* would equal 0 (i.e., perfect equality) if everyone in the population had the same amount of wealth, while *G* would equal 1 (i.e., perfect inequality) if one person in the population had the entirety of the wealth, while everyone else in the population had none. In the water-quality scenarios, *G* = 0 if every moment contributes equally to observed biogeochemical and/or hydrologic responses, whereas *G* = 1 if one moment contributes to the entirety of the observed responses. The appeal of this analysis is its applicability to data over any duration of time or spatial extent in exactly the same mathematical manner.

### 2.3. Temporal Targeting Decision-Making Framework

The results of the temporal inequality analyses were used to develop a decision-making framework to identify the time and flow conditions under which the targeted load was exported. The Lorenz Curves were used to identify the fraction of time during which the cumulative TN and TP loads were exported, respectively, during low-flow and high-flow conditions. The framework then links the targeted loads to a flow-duration curve (FDC) for each site, which enables the flow conditions during the periods of time that the targeted loads were exported to be specified. The decision-making framework produces the specific flowrates for high- and low-flow conditions that export the loads that need to be mitigated to meet the desired load-reduction goals. If the annual load is greater than the targeted load, there are generally two "windows of opportunity" for achieving the load-reduction goals, with low-flow targeting resulting in a longer period of time over which opportunities arise to effectively mitigate the load, while high-flow targeting provides a shorter, more targeted period of time to achieve the same load reduction.

### 3. Results

#### 3.1. Load-Reduction Goals

The percentage of the annual TN and TP loads that must be reduced to meet the load-reduction goals mandated for each county are reported in Table 2. They range from less than 10% of the annual TP load to more than 100% of the annual loads for both TN and TP. For the years in which the load reduction was more than 100% of the annual load, the load was smaller than the mandated load reduction (see Tables 1 and 2), meaning that even if the entire annual load had been effectively mitigated, the annual load reduction would not have been met. Further, the percentage of the annual TN and TP loads that need to be reduced varied across years for each site, with the load reduction needed for some catchments ranging from less than 30% in some years to more than 70% in others (Table 2). The range was particularly large for the West Conewago Creek site, which only needed to reduce its load by approximately 40% in 2011 but, in other years, could have reduced 100% of its load and still not met its annual load-reduction goal (Table 2).

#### 3.2. Temporal Inequality Results

The degree of temporal inequality exhibited by each of the selected study sites was generally lower for TN than TP (Figure 3; Table 3), with an average value of *G* for TN ($G_{TN}$) across all years at all sites of 0.44, while the average *G* for TP ($G_{TP}$) across all years of all sites was 0.67. Across all sites, the range of *G* values exhibited by TN was 0.19 to 0.73, while the range for TP was 0.32 to 0.90. The study site with the lowest degree of temporal inequality for TN and TP was Quittapahilla Creek, which had average $G_{TN}$ and $G_{TP}$ values of 0.24 and 0.41, respectively. The Conewago and West Conewago Creek study sites had the highest average $G_{TN}$ and $G_{TP}$ values of >0.60 and >0.75, respectively.

**Table 2.** Annual loads for total nitrogen (TN) and total phosphorus (TP) for each selected study site, along with the calculated percent reduction of TN and TP needed to meet the annual load-reduction goals provided in Table 1.

| Station ID | Stream Name | Year | Annual Load (kg/y) | | Reduction Needed in Annual Load (%) | |
|---|---|---|---|---|---|---|
| | | | TN | TP | TN | TP |
| 1570000 | Conodoguinet Creek | 2010 | 2,519,351 | 46,921 | 33.65 | 24.38 |
| | | 2011 | 3,020,707 | 84,057 | 28.06 | 13.61 |
| | | 2012 | 2,626,795 | 43,936 | 32.27 | 26.04 |
| | | 2013 | 1,714,809 | 21,728 | 49.44 | 52.65 |
| | | 2014 | 2,257,449 | 39,676 | 37.55 | 28.83 |
| | | 2015 | 1,595,774 | 22,924 | 53.22 | 49.90 |
| | | 2016 | 1,480,014 | 20,259 | 57.28 | 56.46 |
| | | 2017 | 1,329,605 | 16,788 | 63.76 | 68.14 |
| | | 2018 | 3,208,919 | 73,436 | 26.42 | 15.58 |
| 1573160 | Quittapahilla Creek | 2013 | 615,651 | 16,176 | 33.19 | 42.75 |
| | | 2014 | 727,136 | 20,475 | 28.10 | 33.77 |
| | | 2015 | 505,997 | 8,768 | 40.39 | 78.87 |
| | | 2016 | 555,417 | 10,311 | 36.79 | 67.07 |
| | | 2017 | 512,714 | 7,548 | 39.86 | 91.62 |
| | | 2018 | 888,834 | 24,005 | 22.99 | 28.81 |
| 1573695 | Conewago Creek | 2013 | 51,752 | 5,825 | 109.08 | 32.80 |
| | | 2014 | 81,874 | 9,607 | 68.95 | 19.89 |
| | | 2015 | 47,854 | 3,601 | 117.97 | 53.06 |
| | | 2016 | 58,677 | 5,335 | 96.21 | 35.81 |
| | | 2017 | 47,514 | 3,185 | 118.82 | 59.98 |
| | | 2018 | 137,565 | 24,338 | 41.04 | 7.85 |
| 1574000 | West Conewago Creek | 2010 | 1,696,470 | 133,314 | 59.94 | - |
| | | 2011 | 2,683,395 | 357,276 | 37.89 | - |
| | | 2012 | 1,766,520 | 157,856 | 57.56 | - |
| | | 2013 | 1,451,354 | 148,154 | 70.06 | - |
| | | 2014 | 2,017,352 | 225,083 | 50.40 | - |
| | | 2015 | 1,156,909 | 97,761 | 87.89 | - |
| | | 2016 | 1,455,303 | 135,015 | 69.87 | - |
| | | 2017 | 781,830 | 55,144 | 130.05 | - |
| | | 2018 | 2,310,778 | 270,731 | 44.00 | - |
| 1575585 | Codorus Creek | 2013 | 1,320,054 | 77,114 | 40.33 | - |
| | | 2014 | 1,835,336 | 108,968 | 29.00 | - |
| | | 2015 | 950,050 | 41,905 | 56.03 | - |
| | | 2016 | 1,245,531 | 59,141 | 42.74 | - |
| | | 2017 | 740,603 | 31,663 | 71.88 | - |
| | | 2018 | 1,598,243 | 123,154 | 33.31 | - |
| 1576754 | Conestoga River | 2010 | 4,296,086 | 144,816 | 57.82 | 70.06 |
| | | 2011 | 5,284,077 | 431,397 | 47.01 | 23.52 |
| | | 2012 | 3,909,384 | 152,379 | 63.54 | 66.58 |
| | | 2013 | 4,023,552 | 221,673 | 61.73 | 45.77 |
| | | 2014 | 5,027,739 | 298,263 | 49.40 | 34.02 |
| | | 2015 | 3,311,859 | 138,998 | 75.00 | 72.99 |
| | | 2016 | 3,281,919 | 138,687 | 75.68 | 73.16 |
| | | 2017 | 2,473,277 | 87,942 | 100.43 | 115.37 |
| | | 2018 | 4,636,651 | 325,283 | 53.57 | 31.19 |
| 1576787 | Pequea Creek | 2010 | 1,569,087 | 87,343 | 49.85 | 36.58 |
| | | 2011 | 1,482,631 | 120,837 | 52.75 | 26.44 |
| | | 2012 | 1,246,702 | 63,093 | 62.74 | 50.64 |
| | | 2013 | 1,238,306 | 137,569 | 63.16 | 23.22 |
| | | 2014 | 1,605,497 | 146,393 | 48.72 | 21.82 |
| | | 2015 | 1,006,888 | 69,547 | 77.68 | 45.94 |
| | | 2016 | 958,201 | 52,977 | 81.63 | 60.31 |
| | | 2017 | 598,215 | 16,240 | 130.75 | 196.73 |
| | | 2018 | 1,218,037 | 136,156 | 64.21 | 23.47 |

**Table 2.** *Cont.*

| Station ID | Stream Name | Year | Annual Load (kg/y) | | Reduction Needed in Annual Load (%) | |
|---|---|---|---|---|---|---|
| | | | TN | TP | TN | TP |
| 1614500 | Conococheague Creek | 2010 | 2,721,257 | 78,048 | 30.87 | 37.14 |
| | | 2011 | 3,251,720 | 131,625 | 25.83 | 22.02 |
| | | 2012 | 2,839,020 | 64,177 | 29.59 | 45.16 |
| | | 2013 | 2,115,025 | 47,024 | 39.71 | 61.64 |
| | | 2014 | 2,603,312 | 82,177 | 32.27 | 35.27 |
| | | 2015 | 1,511,072 | 32,248 | 55.59 | 89.88 |
| | | 2016 | 1,700,538 | 37,061 | 49.39 | 78.21 |
| | | 2017 | 1,704,557 | 43,349 | 49.28 | 66.86 |
| | | 2018 | 3,647,005 | 137,505 | 23.03 | 21.08 |

-, Mandated load-reduction goal has already been met.

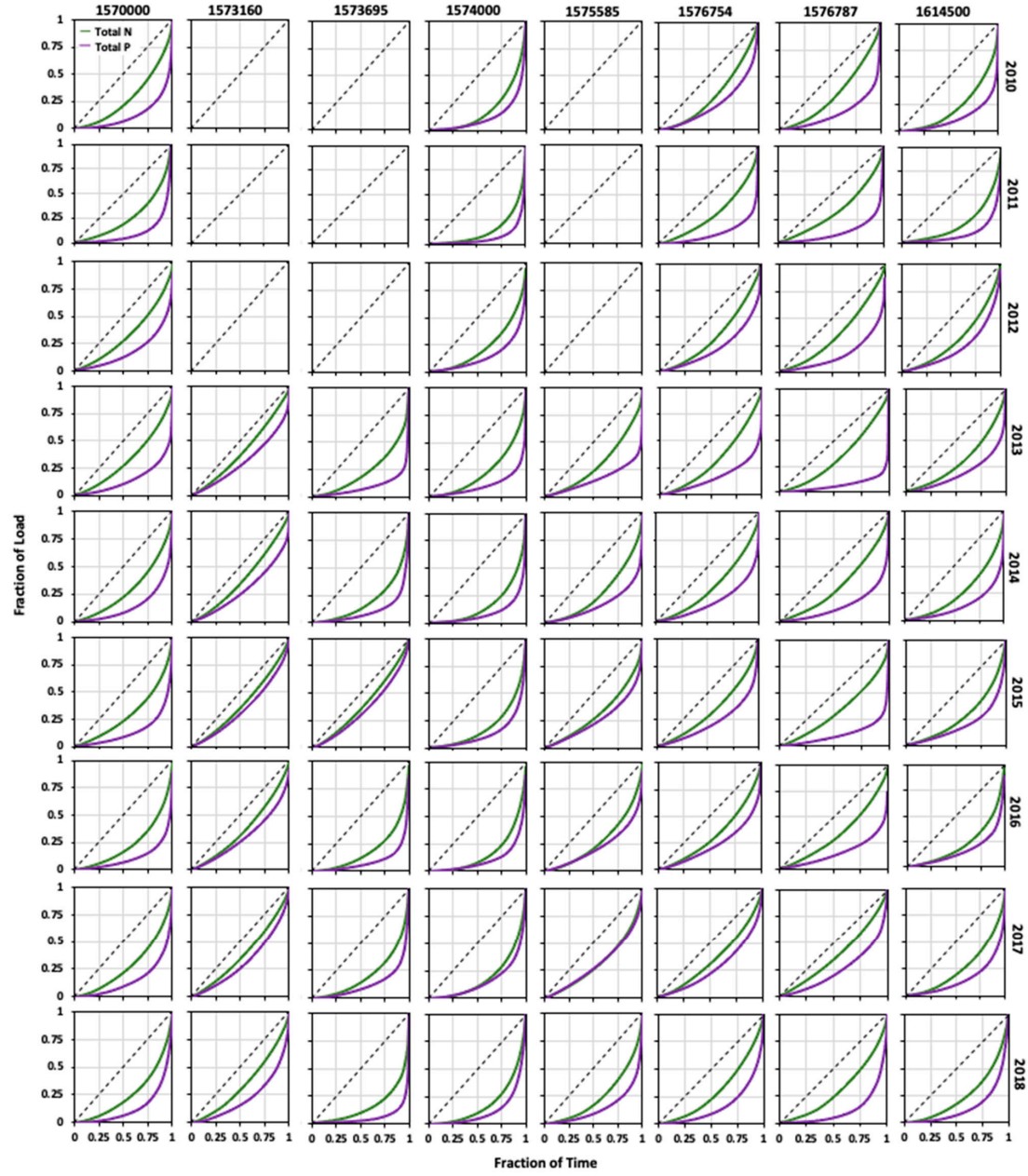

**Figure 3.** Lorenz Curves for total nitrogen (TN) and total phosphorus (TP) loads exported for each year for each selected study site. The dotted line represents the "Line of Equality", with temporal inequality signified by the degree to which the Lorenz Curve plots below the line. Blank plots are shown when no load data are available for those years for specific gauging stations.

**Table 3.** Gini Coefficients for total nitrogen ($G_{TN}$) and total phosphorus ($G_{TP}$) loads exported for each year for each selected study site, along with percentages of time that each load is exported during low- and high-flow conditions.

| Station ID | Stream Name | Year | $G_{TN}$ | $G_{TP}$ | % Time TN | | % Time TP | |
|---|---|---|---|---|---|---|---|---|
| | | | | | Low Flow | High Flow | Low Flow | High Flow |
| 1570000 | Conodoguinet Creek | 2010 | 0.42 | 0.74 | 63.8 | 11.0 | 82.2 | 0.5 |
| | | 2011 | 0.49 | 0.81 | 63.6 | 5.2 | 78.4 | 0.3 |
| | | 2012 | 0.34 | 0.64 | 55.7 | 12.6 | 74.6 | 0.8 |
| | | 2013 | 0.42 | 0.70 | 77.5 | 21.6 | 98.1 | 3.6 |
| | | 2014 | 0.42 | 0.71 | 67.7 | 12.9 | 83.3 | 1.4 |
| | | 2015 | 0.42 | 0.72 | 82.5 | 22.7 | 95.3 | 4.7 |
| | | 2016 | 0.52 | 0.77 | 89.6 | 19.4 | 98.1 | 3.8 |
| | | 2017 | 0.43 | 0.69 | 88.8 | 32.9 | 98.1 | 15.1 |
| | | 2018 | 0.48 | 0.72 | 60.5 | 5.8 | 71.2 | 0.8 |
| 1573160 | Quittapahilla Creek | 2013 | 0.19 | 0.38 | 46.3 | 22.5 | 69.9 | 15.6 |
| | | 2014 | 0.22 | 0.42 | 43.0 | 16.7 | 61.9 | 6.8 |
| | | 2015 | 0.22 | 0.32 | 55.6 | 26.3 | 94.0 | 61.1 |
| | | 2016 | 0.26 | 0.40 | 55.7 | 20.8 | 92.1 | 40.2 |
| | | 2017 | 0.22 | 0.36 | 55.3 | 25.5 | 98.9 | 79.7 |
| | | 2018 | 0.31 | 0.60 | 42.2 | 9.9 | 69.0 | 3.8 |
| 1573695 | Conewago Creek | 2013 | 0.54 | 0.82 | 100.0 | 100.0 | 97.3 | 0.5 |
| | | 2014 | 0.60 | 0.82 | 96.4 | 23.6 | 85.5 | 0.5 |
| | | 2015 | 0.61 | 0.78 | 100.0 | 99.7 | 74.8 | 31.0 |
| | | 2016 | 0.62 | 0.81 | 99.7 | 73.5 | 95.6 | 1.1 |
| | | 2017 | 0.58 | 0.75 | 100.0 | 100.0 | 97.8 | 5.5 |
| | | 2018 | 0.7 | 0.90 | 91.5 | 2.7 | 80.5 | <0.1 |
| 1574000 | West Conewago Creek | 2010 | 0.64 | 0.76 | 92.9 | 16.2 | - | - |
| | | 2011 | 0.73 | 0.86 | 89.3 | 3.3 | - | - |
| | | 2012 | 0.55 | 0.72 | 90.2 | 19.1 | - | - |
| | | 2013 | 0.62 | 0.79 | 97.2 | 25.5 | - | - |
| | | 2014 | 0.68 | 0.82 | 92.1 | 8.2 | - | - |
| | | 2015 | 0.62 | 0.74 | 99.2 | 47.9 | - | - |
| | | 2016 | 0.68 | 0.79 | 96.7 | 19.1 | - | - |
| | | 2017 | 0.63 | 0.69 | 100.0 | 100.0 | - | - |
| | | 2018 | 0.64 | 0.76 | 87.1 | 7.9 | - | - |
| 1575585 | Codorus Creek | 2013 | 0.4 | 0.67 | 68.5 | 15.9 | - | - |
| | | 2014 | 0.46 | 0.67 | 61.1 | 7.1 | - | - |
| | | 2015 | 0.38 | 0.49 | 81.9 | 28.8 | - | - |
| | | 2016 | 0.45 | 0.55 | 75.1 | 13.7 | - | - |
| | | 2017 | 0.36 | 0.37 | 91.5 | 47.7 | - | - |
| | | 2018 | 0.49 | 0.69 | 68.8 | 7.7 | - | - |
| 1576753 | Conestoga River | 2010 | 0.37 | 0.53 | 79.2 | 32.6 | 96.7 | 31.0 |
| | | 2011 | 0.40 | 0.78 | 74.8 | 20.3 | 86.6 | 0.3 |
| | | 2012 | 0.34 | 0.56 | 82.5 | 39.3 | 96.7 | 25.4 |
| | | 2013 | 0.32 | 0.66 | 82.2 | 38.9 | 95.3 | 2.2 |
| | | 2014 | 0.36 | 0.63 | 73.2 | 25.8 | 81.9 | 1.6 |
| | | 2015 | 0.34 | 0.55 | 91.8 | 53.4 | 98.4 | 33.4 |
| | | 2016 | 0.36 | 0.54 | 91.5 | 51.9 | 98.6 | 36.1 |
| | | 2017 | 0.30 | 0.46 | 100.0 | 100.0 | 100.0 | 99.7 |
| | | 2018 | 0.39 | 0.67 | 78.4 | 26.8 | 81.4 | 2.2 |
| 1576787 | Pequea Creek | 2010 | 0.34 | 0.69 | 72.1 | 27.7 | 90.1 | 1.4 |
| | | 2011 | 0.35 | 0.78 | 77.0 | 27.4 | 89.6 | 0.5 |
| | | 2012 | 0.32 | 0.68 | 80.6 | 40.2 | 95.4 | 5.2 |
| | | 2013 | 0.32 | 0.85 | 84.4 | 41.6 | 96.7 | <0.1 |
| | | 2014 | 0.35 | 0.72 | 72.1 | 25.8 | 76.2 | 0.3 |
| | | 2015 | 0.33 | 0.77 | 94.2 | 58.1 | 98.4 | 1.1 |
| | | 2016 | 0.33 | 0.69 | 94.3 | 62.8 | 99.5 | 6.6 |
| | | 2017 | 0.24 | 0.51 | 100.0 | 100.0 | 100.0 | 99.7 |
| | | 2018 | 0.39 | 0.76 | 86.6 | 37.0 | 83.0 | 1.1 |
| 1614500 | Conococheague Creek | 2010 | 0.54 | 0.75 | 69.9 | 5.5 | 93.7 | 1.1 |
| | | 2011 | 0.60 | 0.81 | 71.0 | 3.6 | 87.1 | 0.5 |
| | | 2012 | 0.40 | 0.56 | 58.2 | 10.1 | 84.7 | 10.1 |
| | | 2013 | 0.46 | 0.63 | 71.8 | 12.3 | 98.1 | 13.2 |
| | | 2014 | 0.48 | 0.72 | 66.3 | 7.7 | 90.4 | 0.8 |
| | | 2015 | 0.46 | 0.62 | 85.5 | 22.5 | 100.0 | 60.0 |
| | | 2016 | 0.55 | 0.68 | 87.2 | 12.0 | 99.2 | 26.0 |
| | | 2017 | 0.48 | 0.68 | 81.6 | 17.5 | 98.1 | 14.5 |
| | | 2018 | 0.50 | 0.71 | 57.8 | 4.7 | 76.7 | 1.6 |

-, Mandated load-reduction goal has already been met; <0.1%, targeted load exported in a single day.

In general, sites with a higher percentage of agricultural and developed area had lower values of $G_{TN}$ and $G_{TP}$. The Conodoguinet Creek drainage area comprises approximately 15% forested land, with nearly 85% of the drainage area either developed or agricultural land use (Table 1). Conodoguinet Creek's average values of $G_{TN}$ and $G_{TP}$ were 0.24 and

0.41, respectively, which were the lowest of all the selected sites (Table 3). West Conewago Creek, which was the site with the highest percentage of forested land use (Table 1), had the highest average value of $G_{TN}$ and the second highest average value of $G_{TP}$ (0.64 and 0.77, respectively; Table 3).

*3.3. Decision-Making Framework Results*

The decision-making framework links annual load-reduction goals to the specific percentage of days within the year needed to fully mitigate the targeted load under either high- or low-flow conditions (Figure 4; Table 3). Thus, through the framework, a watershed planner can identify two "windows of opportunity" in which to mitigate exported loads sufficiently to meet the annual reduction goals for a particular pollutant. To illustrate the framework's utility, data from 2014 at the Conococheague Creek site were selected and analyzed to determine the specific periods of time and corresponding flow conditions during which the targeted loads were exported. The framework shows that to achieve a 32% TN load-reduction goal, either 66.3% of low-flow conditions or 7.7% of high-flow conditions must be targeted. Based on the flow-duration curve for the site, the flowrates when those loads were exported were less than 20 m³/s if lower flowrates were targeted for treatment. However, if high flowrates were targeted, loads exported were greater than 42 m³/s (Figure 4). The extent of temporal inequality exhibited by the TP loads meant that either nearly all flow conditions (i.e., flowrates observed less than 90.4% of the time) needed to be targeted to achieve the 35% annual load-reduction goal, or the highest 0.8% of flow conditions (i.e., flowrates higher than 150 m³/s; Figure 4) could be targeted and achieve the same load-reduction goals.

The temporal targeting analyses across all sites and all years reveal TN and TP loads equivalent to the mandated load-reduction goals can be exported within as little 2.7% of the year for TN and < 0.5% for TP. These results suggest that mandated load-reduction goals could sometimes be achieved by effectively targeting loads exported over less than ten days of the year. In several cases, the effects of a single storm event were so high that the targeted load under high-flow conditions was exported in a single day (Table 3). The effects of these extreme events on the shape of the Lorenz Curve can be seen for Conewago Creek in 2018 and Pequea Creek in 2013 (Figure 3). However, mitigating the loads of such extreme events requires similarly extreme conservation practices that are designed far beyond those intended for everyday mitigation.

Conversely, during lower flow conditions, temporal targeting results showed that loads equivalent to the annual load-reduction goals were never exported less than 42% of the time for TN or less than 62% of the time for TP across any of the sites (Table 3). To mitigate TN and TP loads during low-flow conditions, conservation practices or best management practices need to be effective in treating TN and TP loads over longer stretches of consecutive days during and between small storm events. However, conservation practice effectiveness depends not only on storm patterns but also on crop-rotation cycles and land-use management. Thus, evaluating seasonal patterns of time-series graphs across several historical years in context of a specific watershed's typical cropping and land-cover patterns may help estimate expected effectiveness and lifecycles of long-term agricultural conservation practices.

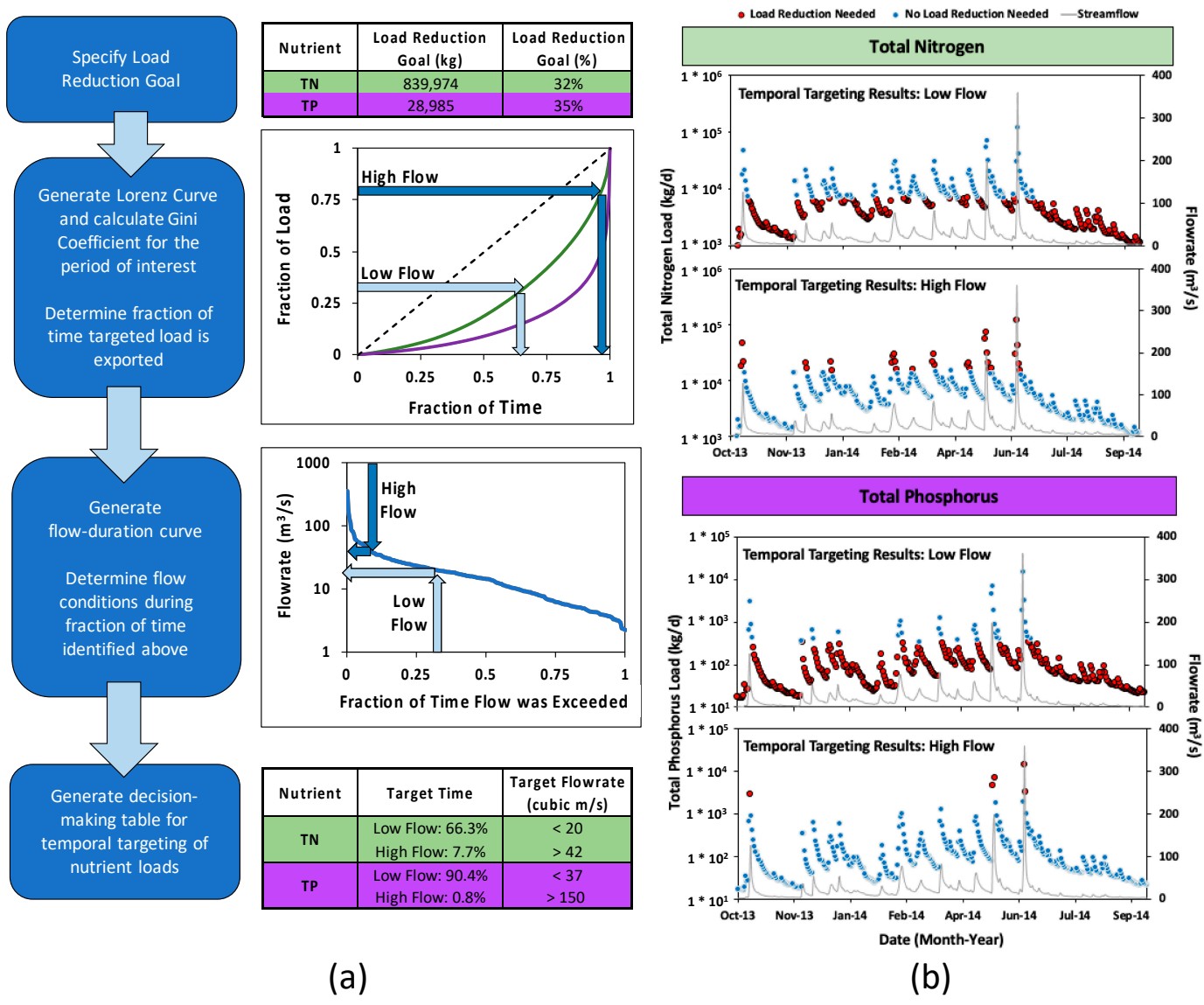

**Figure 4.** (**a**) Decision-making flow chart demonstrated for Conococheague Creek (2014) to determine site-specific fractions of time during which the targeted loads are exported and identify the corresponding flow conditions; arrows on the Lorenz Curve and flow-duration curve are shown for targeting total nitrogen (TN) load. (**b**) Time-series graphs highlight specific events during which targeted loads for TN and total phosphorus (TP) are exported during low- and high-flow conditions.

## 4. Discussion

This novel application of Lorenz Inequality and corresponding *G* demonstrates the utility of leveraging an analysis commonly used in economics for quantifying income inequality for quantifying the temporal inequality of contaminant loads and establishing a framework for temporal targeting of "hot moments" to achieve load-reduction goals. Earlier efforts to apply Lorenz Inequality to hydrology and water-quality data have helped to explain the effects of scale (both temporal and spatial) on the degree of temporal inequality exhibited by discharge and loads, with higher inequality in smaller headwater catchments and lower inequality in larger watersheds as well as higher inequality when finer temporal resolution data (e.g., daily scale) are used to generate the Lorenz Curves compared to coarser temporal scales (e.g., monthly) [11]. Further, across the Chesapeake Bay, a wide range of temporal inequality was documented for 108 stations in the Chesapeake Bay Nontidal Network, with *G* ranging from 0.24–0.60 for flow, 0.18–0.69 for TN, 0.36–0.92 for

TP, and 0.39–0.90 for total suspended sediment [15]. However, the results of the previous analysis across the Chesapeake Bay watershed were for the entire 2010–2018 period since the TMDL was enacted, limiting the utility of those results for decision making at the annual scale.

Here, the results of this temporal inequality analysis demonstrate the potential effectiveness of targeting "hot moments" to achieve load-reduction goals in impaired surface water bodies. For catchments with a high degree of temporal inequality (i.e., G approaching 1), this temporal targeting is especially important, as failing to adopt conservation practices that do not adequately reduce loads during high-flow conditions may prevent load-reduction goals from being met. Conversely, in catchments with low degrees of temporal inequality, the period of time over which the targeted load is exported is longer, and spatial targeting may be more effective than temporal targeting for meeting load-reduction goals, with more opportunities available to effectively reduce the load over the course of a year. The implications of this analysis may be helpful in understanding difficulties in meeting water-quality-restoration goals in long-impaired watersheds, such as the Chesapeake Bay.

The results demonstrate that while in some years, load-reduction goals are only a small portion of the overall load exported and may easily be met by targeting a few storm events, in other years, loads are actually less than the targeted load, and even if 100% of the annual load were effectively mitigated, the annual load-reduction goal could not be met. Viewing these expectations through the lens of temporal inequality can be helpful in understanding how easy or difficult achieving load-reduction goals will be in a given watershed since the higher the value of G, the more difficult it will be to reduce loads without capturing and treating high-flow conditions. When the G is relatively low, as is often the case for TN [15], it is because export of the constituent of interest occurs largely during baseflow conditions, and therefore, achieving load reduction without effectively treating high-flow conditions may be possible. In these cases, conservation practices that help reduce groundwater concentrations, such as cover crops and other nonstructural best management practices (BMPs), may be most effective in meeting load-reduction goals. However, when G is high, as is often the case for TP [15], it is because export of the constituent of interest largely occurs during high-flow conditions, and therefore, if these events are not effectively treated, meeting the load-reduction goal may not be possible. In these cases, conservation practices, such as riparian buffers, vegetated filter strips, and detention basins, may be most effective in meeting the load-reduction goals. Water-quality BMPs are often vegetative and therefore are mainly effective in managing low-flow events over only the portion of the year when the plants are actively growing and are not dormant. The portions of the year during which water-quality BMPs are effective should align with months during which large portions of annual loads are exported. This may require landowners to consider designing BMPs to manage high-flow events or a combination of low-flow BMPs that are effective over a longer time interval.

## 5. Conclusions

Overall, the results of this research demonstrated a decision-making framework that can be applied at any temporal or spatial scale to quantify the importance of targeting "hot moments" to achieve specific load-reduction goals. The results of our analysis demonstrate the site-specific nature of the results such that even across a watershed with a single TMDL, the implementation of conservation practices that will achieve the load-reduction goals is likely to be heterogeneous, with the success of field-scale implementation of appropriate conservation practices relying on local knowledge of hydrology and contaminant transport.

**Author Contributions:** Conceptualization, H.E.P., T.L.V., and M.R.; methodology, H.E.P. and T.L.V.; software, N.O., D.S., and T.L.V.; formal analysis, N.O. and D.S.; investigation, N.O., D.S., H.E.P., and T.L.V.; resources, H.E.P.; writing—original draft preparation, N.O., D.S., and H.E.P.; writing—review and editing, T.L.V. and M.R.; visualization, N.O., D.S., H.E.P., and T.L.V.; supervision, H.E.P. and M.R.; project administration, H.E.P.; funding acquisition, H.E.P. All authors have read and agreed to the published version of the manuscript.



**Funding:** Nicole Opalinski worked on this project as part of an independent studies course. Daniel Schultz was funded through support from the Penn State College of Engineering Summer Research Experience for Undergraduates Program and from the Penn State Erickson Discovery Undergraduate Research Program. Heather E. Preisendanz is funded, in part, by the Penn State Institutes of Energy and the Environment and by the USDA National Institute of Food and Agriculture Federal Appropriations under Project PEN04574 and Accession number 1004448. Matt Royer was supported, in part, by the Penn State Center for Nutrient Solutions, which is funded by an EPA STAR Grant (RD835568). All entities involved are equal opportunity providers and employers.

**Institutional Review Board Statement:** Not applicable.

**Informed Consent Statement:** Not applicable.

**Data Availability Statement:** All data are publicly available through the United States Geological Survey and can be accessed at: https://cbrim.er.usgs.gov/introduction.html or https://doi.org/10.5066/P931M7FT.

**Conflicts of Interest:** Authors Preisendanz and Veith reside in the same household. The funders had no role in the design of the study; in the collection, analyses, or interpretation of data; in the writing of the manuscript, or in the decision to publish the results.

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
