# Peer review of "Meeting the Moment: Leveraging Temporal Inequality for Temporal Targeting to Achieve Water-Quality Load-Reduction Goals"

_water, doi:10.3390/w14071003_

Round 1
Reviewer 1 Report
The paper addresses the interesting and relevant implementation of Lorenz Inequality decision-making framework that uses Lorenz Curves and Gini Coefficients to quantify the degree of temporal inequality exhibited by contaminant loads and demonstrate its utility for eight impaired catchments in the Chesapeake Bay watershed. The paper has a nice presentation, clear to read, and well referenced. Congratulations to the authors. I enjoyed reading your paper. I recommend the paper to be published with considering the following comment:
I couldn't find "Conclusion" section in the paper. Please follow the journal's guidelines and add this section to the manuscript if it is mandatory for research papers.
Good luck
Reviewer 2 Report
The manuscript titled „Meeting the moment: Leveraging temporal inequality for temporal targeting to achieve water quality load reduction goals” is interesting and valuable, describes important aspects. The article has been prepared very carefully, the material is presented appropriately and clearly but I suggest two optional correction before publishing.
1) change the scale of the maps in Figure 1 to make them larger and more readable,
2) change the structure of the article so that there are conclusions from the interpretation at the end and section titled "Conclusions".
